# A Comprehensive Review on Wastewater Nitrogen Removal and Its Recovery Processes

**DOI:** 10.3390/ijerph20043429

**Published:** 2023-02-15

**Authors:** Yifan Zhou, Yingying Zhu, Jinyuan Zhu, Chaoran Li, Geng Chen

**Affiliations:** Faculty of Maritime and Transportation, Ningbo University, Ningbo 315211, China

**Keywords:** wastewater, nitrogen removal, environment, waste treatment, nitrogen recovery

## Abstract

Discharging large amounts of domestic and industrial wastewater drastically increases the reactive nitrogen content in aquatic ecosystems, which causes severe ecological stress and biodiversity loss. This paper reviews three common types of denitrification processes, including physical, chemical, and biological processes, and mainly focuses on the membrane technology for nitrogen recovery. The applicable conditions and effects of various treatment methods, as well as the advantages, disadvantages, and influencing factors of membrane technologies, are summarized. Finally, it is proposed that developing effective combinations of different treatment methods and researching new processes with high efficiency, economy, and energy savings, such as microbial fuel cells and anaerobic osmotic membrane bioreactors, are the research and development directions of wastewater treatment processes.

## 1. Introduction

Nitrogen (N) is essential for proteins, nucleic acids, enzymes, denitrifying bacteria, nitrifying bacteria, and ammoniating bacteria [1]. It is indispensable for plant growth, photosynthesis, energy transfer, and fertilizer synthesis [2]. However, excess nitrogen can harm aquatic ecosystems, causing eutrophication of amphibian systems and the depletion of dissolved oxygen, which could lead to the death of fish and other marine organisms [3]. In addition, nitrate leaching from soil can contaminate groundwater, which, when ingested, could cause methemoglobinemia in infants under three months [4]. Since industrial sewage, agricultural sewage, and domestic sewage often contain various nitrogen-containing pollutants (such as ammoniacal nitrogen, nitrite nitrogen, nitrate nitrogen, etc.) [5], denitrification of wastewater is essential to curbing indiscriminate discharge of sewage below the required standard.

So far, there have been several reviews on nitrogen removal from wastewater. Mishra et al. [6] reviewed the effectiveness and complexity of the biological nitrogen removal process and discussed the key operating parameters and bioreactor configurations for nitrogen removal. Yellezuome et al. [7] reviewed the advantages and disadvantages of ammonia stripping technology and discussed in depth the influence of the anaerobic digestion process and stripping parameters on ammonia removal efficiency. Karri et al. [8] reviewed the technologies implemented for the removal of ammonia and its related compounds from wastewater and the effects of process parameters on the efficiency of the adsorption process.

Nowadays, the main wastewater nitrogen removal methods can be divided into physical, chemical, and biological processes [9]. Physical treatment methods, such as ammonia stripping, ion exchange, and adsorption [10,11,12], have high retention rates for organic pollutants [13]. Still, these methods are expensive and cause secondary pollution that requires further treatment [14]. Chemical processes (based on breakpoint chlorination and magnesium ammonium phosphate hexahydrate (MAP) precipitation) [15,16,17] have the advantages of simple operation, fast reaction speed, and high-speed denitrification [18]. However, chemical methods are labor-intensive, expensive, and require secondary processing [19]. Through microbial actions, the biological processes convert organic nitrogen and ammoniacal nitrogen in the sewage to nitrates and nitrites, which are finally converted to nitrogen through ammonization, nitrification, and denitrification [20]. The biological option is mature, with a high pollutant degradation efficiency and no secondary pollution. However, the process is tedious, often requiring additional carbon sources and consuming a significant amount of energy [21].

The continuous improvement of detection technology, especially the rise of on-site rapid detection technology [22,23], has made the detection of ammonia nitrogen concentration more efficient, executable, faster to operate, and more cost-effective. These characteristics offer enormous potential for the early and precise detection and assessment of ammonia nitrogen concentration in wastewater, which is crucial for human health and the global safety of environmental water [24,25]. At the same time, increasingly stringent discharge requirements for nitrogen content have prompted extensive research to improve denitrification efficiency and performance [6,26]. A search on “ScienceDirect” with the keywords “wastewater” and “remove nitrogen” found that more than 70,000 articles on wastewater denitrification have been published in the last decade (2012–2021), with the number of published articles increasing with time. Most of these articles are published in journals in environmental science, chemical engineering, and energy ((a)). Figure 1b shows the degree of attention given by scholars to various denitrification methods.

With the diverse applications of nitrogen, nitrogen demand is gradually increasing. Many nitrogen resources exist in water and soil as typical pollutants after use [27]. Therefore, nitrogen recovery is crucial to the circular economy and environmental remediation. At present, several reviews on nitrogen recovery have been published. Qin et al. [28] emphasized the importance of nitrogen recovery and introduced several nitrogen purification technologies in domestic sewage treatment, including excess activated sludge concentration, ion exchange and adsorption, electrodialysis, etc., and evaluated the advances and compatibilities of N up-concentration technologies. Rahimi et al. [29] discussed the basic biological denitrification processes in wastewater treatment, including nitrification, denitrification, and anammox, as well as the influencing physicochemical factors, advantages, and challenges of these methods. However, these technologies still have defects, such as the fact that the biological method is only suitable for low ammonia nitrogen wastewater and is time-consuming; the chemical method is only suitable for high ammonia nitrogen wastewater and will cause secondary pollution due to the chemical added; and the physical method is less effective in recovering nitrogen [19,30,31]. Therefore, it is challenging to increase the nitrogen recovery efficiency from various concentrated wastewater while meeting the discharge standard. 

A promising approach to achieving these two conditions is membrane technology [32,33]. Due to the peculiar selective separation feature of the membrane, NH_3_ can pass through the membrane pores to the acidic solution, where NH_3_ combines with free protons to form non-volatile ammonium cations (NH_4^+^_). Then, these cations are converted into valuable ammonium salts for fertilizer [34]. The nitrogen removal by membrane separation has attracted researchers’ attention. A comprehensive review of various membrane separation processes for nitrogen recovery from waste water is given by Al-Juboori et al. [35]. In addition to this, the existing conceptual designs of some membrane processes were carefully reviewed, and new designs were proposed for more efficient and resilient processes in this view. However, current review articles focus more on various biological removal and recovery technologies as well as their energy costs [29], microbiological characteristics analysis, key operating parameter identification, bioreactor design and process control strategy optimization [36,37,38,39], and biochemical removal process development [40]. To the best of the authors’ knowledge, there is a lack of systematic reviews of wastewater denitrification and nitrogen recovery. Therefore, this paper reviews standard wastewater denitrification and nitrogen recovery processes in recent years. It focuses on the factors affecting each method, the advantages and disadvantages of each operation in wastewater treatment, and discusses the current problems in denitrification and nitrogen recovery. This paper also includes future developmental trends.

## 2. Nitrogen Removal Process

Current NH_3_/NH_4^+^_ removal methods have environmental and technical advantages and limitations. Table 1 summarizes several NH_3_/NH_4^+^_ ion removal techniques, including chemical precipitation, adsorption, and biological processes.

### 2.1. Chemical Precipitation

Chemical precipitation in wastewater treatment transforms dissolved unwanted substances into solid particles [46]. This method is popular in treating domestic sewage, industrial wastewater, pig farm wastewater, landfill leachate, etc. [47,48]. Specifically, forming MAP precipitate by adding magnesium salts and phosphates effectively removes ammoniacal nitrogen [49]. Figure 2 depicts the general steps of the MAP precipitation method.

Magnesium ammonium phosphate is also called struvite in the waste water treatment industry. Struvite is a white crystal compound that commonly accumulates in post-anaerobic digestion pipes in wastewater treatment plants (WWTPs) [50]. Struvite is formed when magnesium, ammonium, and phosphate that are in equimolar concentrations in waste water [51,52]; the chemical reaction equation is as follows (Equation (1)):Mg^2+^ + NH_4^+^_ + PO_4_^3−^ + 6H_2_O⇋MgNH_4_PO_4_·6H_2_O↓(1)
S = 0.169 (g/L)(2)

The H_2_O solubility of struvite is extremely low, e.g., 0.169 g/L at 25 °C, as shown in Equation (2), thereby enabling an effective way of NH_4^+^_ removal by separation of the struvite precipitate from the aqueous phase. As the precipitation of struvite requires equal molar ratios between the reactants (Equation (1)), if one or more reactants are deficient, the supply of the missing ion is required [53]. In most cases, Mg^2+^ concentrations are generally lower than NH_4^+^_ and PO_4_^3−^ [54]. Thus, a Mg source (MgCl_2_ or MgO) should be added. Compared with other methods, chemical precipitation is economical and straightforward, and its product is comparable to commercial fertilizers. Also, it can be used to precipitate fertilizers and garden soils [55].

#### 2.1.1. Factors Influencing Chemical Precipitation

Several variables influencing chemical precipitation in wastewater treatment include pH, Mg^2+^:NH_4^+^_:PO_4_^3−^, and the Mg source [56]. Studies [57] have shown that MAP precipitates between pH 7 and11. Crystallization occurs at pH 8–9, confirming the effect of pH on NH_4^+^_ removal [58,59]. All these studies indicate a pH range from 7.0 to 11.5 as the optimum for NH_4^+^_ removal. 

Hu et al. [60] studied the effect of pH value on the removal efficiency of ammonia nitrogen by the map precipitation method, and the results showed that when the pH increased from 8 to 9.5, the NH_4^+^_ removal efficiency increased rapidly from 40% to 85.9%. However, when the pH continued to increase, the removal efficiency gradually decreased. Similarly, Zhang et al. [61] used the same method to remove ammonium nitrogen from coking wastewater. They observed that when the pH was increased from 8.5 to 9.5, the nitrogen removal increased from 52% to 81%. However, when the pH value was >9.5, Mg_3_(PO_4_)_2_ was generated instead of MAP, hindering ammonium nitrogen removal. Through electron microscopy, Moulessehoul et al. [62] observed that only MAP crystallizes between pH 8.5 and 11.8. The crystals directly relate to the saturation state and the precipitation pH. They also observed that when the pH was lowered from 11.8 to 8.5, the crystallization increased from 75% to 98%. Elsewhere, Jia [63] assessed MAP precipitates under different pH conditions in synthetic wastewater. The optimal pH range was 9.0–9.5.

Studies [64] have shown that the initial ammoniacal nitrogen concentration influences MAP crystals’ purity. The higher the excess NH_4^+^_ relative to the theoretical ratio of Mg^2+^ and PO_4_^3−^ (1:1:1), the higher the purity of the MAP crystals. However, ammoniacal nitrogen removal is significantly reduced. According to the “same ion effect” [65], increasing the ratio of Mg^2+^ and PO_4_^3^ promotes the forward reaction and improves denitrification efficiency. By changing the [Mg^2+^]:[NH_4^+^_]:[PO_4_^3−^] ratio from 1:1:1 to 2.5:1:1, Di Iaconi et al. [66] enhanced the NH_4^+^_ removal efficiency of a MAP precipitation method from 67% to 95%. Likewise, Türker et al. [67] increased the [Mg^2+^]:[NH_4^+^_]:[PO_4_^3−^] to 1.2:1:1.2 at a pH of 8.5 to obtain a >95% ammoniacal nitrogen removal. 

The Mg sources used for MAP precipitation include Mg (OH)_2_, MgO, MgCl_2_, etc. Due to its excellent water solubility and rapid reaction rate, MgCl_2_ is often used with Na_2_HPO_4_ for NH_4^+^_ precipitation from wastewater. Ammoniacal nitrogen is an ideal precipitant [68]. Guan et al. [69] used MAP precipitation to treat pig wastewater while comparing three Mg sources: MgCl_2_, Mg(OH)_2_, and MgO. They reported that when the molar ratio of MgCl_2_:P was 4:1, large amounts of P, Fe, and Cu could be removed after 150 min of treatment. In addition, the trace metal content of the MAP crystals was low. The removal of P and Fe was significantly less than when the other two magnesium sources, Mg (OH)_2_ and MgO, were used. Since the acidity of MgCl_2_ hydrolysis reduces the solution pH, increasing the amount of adjusted alkali would reduce the treatment cost. Therefore, in recent years, some researchers have proposed using seawater from coastal areas and brine (mainly containing MgCl_2_) remaining after salt production instead of MgCl_2_ industrial products to achieve specifically desirable results [70].

#### 2.1.2. Disadvantages of Chemical Precipitation

Although chemical precipitation has low cost and ease of operation advantages, its high operating costs are due to the use of chemicals, high energy costs, and sludge disposal [71]. In addition, due to the limitation of the solubility product, after the ammoniacal nitrogen in the wastewater reaches a specific concentration, a reagent is added, which lowers the removal effect, resulting in a significant increase in the input cost. Therefore, the chemical precipitation method must be used in other ways suitable for advanced treatment. In recent years, MAP precipitation has been combined with adsorption [72]. In particular, an adsorbent with Mg^2+^ can exchange with NH_4^+^_ to form the MAP precipitate.

### 2.2. Adsorption Method

Adsorption is widely used due to its flexibility in design and operation, high efficiency in pollution control, reusability of adsorbents, low environmental impact, and low cost of adsorbents. Adsorptive denitrification is a mass transfer process where the hydrophobic properties of non-ionic ammonia, and its high affinity for solid surfaces, continuously propel its movement from water to solid surface [73], where it is fixed through physicochemical interactions (Figure 3).

An adsorption phenomenon could be physical, chemical, or both. Physical adsorption occurs when the attractive force between the adsorbate and the adsorbent is a van der Waals force. This adsorption type can be reversed by increasing the temperature or lowering the pressure. Wei et al. [74] used lanthanum-modified zeolite to reduce 92% of the total nitrogen release in sediments. Zhao et al. [75] prepared a novel hollow NiO/Co@C magnetic nanocomposite derived from a metal-organic framework for adsorbing organonitrogen pesticides. The NiO/Co@C exhibited efficient solid-liquid separation and excellent repeatability and stability.

On the other hand, chemisorption assumes that the existing attractive forces and chemical bonds between the adsorbate and the adsorbent are somewhat equal in strength [76]. This adsorption type is irreversible since the constituents involved are much more robust and have some degree of electron transfer (such as chemical bond formation) [77]. For example, Yunnen et al. [78] used D113 ion exchange resin to remove low- and medium-concentration ammoniacal nitrogen from wastewater. They reported that adsorption lowered the concentration of rare earth metals in the initial rare earth metallurgical wastewater from 116 to 13 mg/L.

#### 2.2.1. Factors Influencing Adsorption

pH is an essential factor that affects adsorption significantly. Under acidic conditions, H^+^ competes with NH_4^+^_, hindering NH_4^+^_ collection on the adsorbent [79]. In alkaline environments, there are fewer NH_4^+^_, decreasing the removal efficiency of ammoniacal nitrogen with the pH increase [80]. Therefore, NH_4^+^_ adsorption is most favorable under neutral pH conditions [81,82]. Liu et al. [83] synthesized a zeolite with fluidized fly ash. As the pH increased from 2, NH_4^+^_ adsorption increased, reaching the optimum at pH 6 before the adsorption capacity decreased steadily to pH 8. Elsewhere, Cheng et al. [84] determined the optimal pH for NH_4^+^_-N removal. They investigated the ion-exchange performance of a modified zeolite at different pHs. The modified zeolite had the optimum removal efficiency for NH_4^+^_-N (92.13%) at pH 8. 

Also, the contact time and surface ratio between an adsorbent and its adsorbate are other essential factors affecting NH_4^+^_ removal. In general, removal efficiency gradually decreases with contact time as the initially vacant adsorption sites are occupied by NH_4^+^_, whose concentration gradient is high. As the adsorption sites and concentration gradients decrease, the NH_4^+^_ adsorption rate also decreases [85]. In another study, Chen et al. [86] compared the adsorption rates of NH_4^+^_ on rice husk silica, silicon carbon composites, and activated carbon. They found that the higher the specific surface area, the higher the adsorption efficiency. The adsorbent dosage also controls the adsorption equilibrium, i.e., NH_4^+^_ removal increases with adsorbent dosage. This trend is attributed to the increased adsorbent surface area and the number of active surface sites from the high adsorbent dosage [87]. 

Similarly, El-Shafey et al. [88] studied NH_4^+^_ adsorption on natural and modified Egyptian kaolinite. They found that NH_4^+^_ removal was optimized with 0.1 g of adsorbent dosage. When the concentration increased (>0.1 g), the adsorption decreased, probably due to active site interferences on the adsorbent. In addition, wastewater usually contains Na^+^, K^+^, Mg^2+^, Ca^2+^, etc., which may compete with NH_4^+^_ adsorption for available ion exchange sites, thereby lowering the NH_4^+^_ removal capacity of the resin [89]. Mazloomi and Jalali [90] studied the NH_4^+^_ removal from aqueous solutions using Iranian natural zeolites in the presence of organic acids, cations, and anions. The presence of K^+^ Ca^2+^, Mg^2+^, Cl^−^, PO_4_^3−^, and SO_4_^2−^ in the solution interfered, and thus reduced NH_4^+^_ adsorption.

#### 2.2.2. Disadvantages of Adsorption

Complex adsorbent regeneration, the need for a specific pH range, and the insufficient adsorption capacity of common adsorbents (such as activated carbons) limit adsorption. As an improvement, Liao et al. [91] combined activated carbon with zeolite to remove NH_4^+^_ more effectively by taking advantage of their benefits. The results showed a synergistic effect between the two sorbents, improving NH_4^+^_ removal. Furthermore, adsorbents have a short life span and require constant replacement, generating wastes that require further processing at extra cost [92]. Recently, some studies [93] have used composite technology to graft N-functional groups on graphene, improving the sorbent’s adsorption performance. Steam activation, acid/base activation, metal-salt impregnation, and oxidant treatment are other surface modification techniques commonly used to engineer an adsorbent’s physical and chemical properties, especially activated carbons.

### 2.3. Biological Nutrient Removal (BNR)

The biological method removes nitrogen through natural microbial metabolism. As the most common technology in sewage treatment, it is economical, effective, easy to operate, and generates no secondary pollution. Conventional nitrification-denitrification processes generally achieve physical nitrogen removal from wastewater [94]. Figure 4 illustrates the steps involved in physical denitrification.

However, current operations based on this process are considered economically and environmentally unsustainable because of energy consumption, sludge production, and greenhouse gas emissions. In recent decades, scholars have been trying to find more efficient alternatives to conventional methods to achieve sustainable development [95]. Table 2 compares the total denitrification efficiency, cost, effect, and main technical parameters of various denitrification processes with conventional techniques.

#### 2.3.1. Factors Influencing the Biological Method

The BNR performance primarily depends on microbial growth and activity. In turn, microbial development is influenced by pH, temperature, sludge retention time (SRT), carbon to nitrogen ratio (C/N), hydraulic retention time (HRT), dissolved oxygen (DO), and competition within the microbial diversity. Guo et al. [102] used the RSM model to give the effect of pH value on the removal of TN by strain HNR. The results showed that the TN removal efficiency decreased sharply when the pH was higher than 8.75 or lower than 6.25, and the best TN removal efficiency was reached at pH 8. It indicated that relatively strong alkaline or acidic conditions may inhibit the aerobic denitrification process. Also, Zhang et al. [103] studied the effect of pH on the denitrification of microbial fuel cells. After 40 h of reaction, the conversion rate of cathodic NH_4^+^_ gradually decreased with the pH from 7–8.5; it was consistent with the increase in anolyte pH. This trend indicates that the DO concentration during nitrosation and nitrification mainly drives the catholyte’s pH change. 

DO is another critical factor affecting the BNR process. The DO requirements in traditional nitrification and denitrification processes range from 1.0 to 7.0 mg/L. Under fully aerobic operating conditions, the optimum DO requirement is 6.0–7.0 mg/L, but 1.0–4.0 mg/L under anoxic-aerobic conditions [104]. Zhang et al. [103] studied changes in NH_4^+^_, NO_2^−^_, and NO_3^−^_ concentrations in the reactor under different cathodic DO values. The results showed that NH_4^+^_ removal and nitrification activity were higher at higher DO concentrations. Whereas the DO decreased from 5 to 6.8, the NH4^+^ removal rate showed a downward trend but changed little when the DO was lower, indicating that at high DO concentrations, anodic microbes are more active, consuming some NH_4^+^_; i.e., the more the DO, the less the NO_2^−^_ accumulated in the cathode chamber. Lei et al. [105] showed that low DO can effectively inhibit the growth of nitrite-oxidizing bacteria (NOB), causing nitrite accumulation. At the same time, when the DO concentration is low, the conversion rate of ammoniacal nitrogen is relatively slow.

The medium temperature significantly affects the activity of the bacteria involved in the BNR process. Daverey et al. [106] studied the ambient temperature of swine wastewater treated with the simultaneous partial nitrification, anammox, and denitrification (SNAD) process. They reported that the BNR efficiency of the system is <50% when the winter temperature varies between 15 ºC and 20 °C. At 27 °C, the average NH_4^+^_ removal and the overall BNR efficiencies of the system were 93% and 79%, respectively. In contrast, at >25 °C, the growth and activity of the ammonium oxidizing bacteria (AOB) increased rapidly, promoting nitrite accumulation. 

In addition, denitrification is highly dependent on C/N. A high C/N can denitrify all the nitrates formed during nitrification [107]. Machat [108] studied the effect of C/N on the biological nitrogen removal of an integrated fixed membrane-activated sludge reactor for nearly half a year. The results showed that when the C/N was ten, the nitrogen removal efficiency was optimum, and the average removal of NH_4^+^_ and total nitrogen were 96.54% and 86.1%, respectively. On the contrary, when C/N was four, the removal rate decreased significantly, and the respective average removal efficiencies reduced to 82% and 53%. 

The bacterial growth and activity in BNR bioreactors can also be affected by the HRT of the wastewater. Chen et al. [109] studied the effect of HRT on the BNR efficiency and microbial diversity of the SNAD process for wastewater treatment in a packed bed reactor. They found that when the HRT was reduced from 18 to 3 h, the system efficiency dropped significantly from 84% to 38%.

#### 2.3.2. Disadvantages of the Biological Method

Microorganisms and the sludge mainly influence the processes involved in the BNR process. In addition, there is a need for a high level of operator competence and strict periodic inspection. Complex sludge settling capacity requires large reactors, settling tanks, high operating costs, slow reaction rates, NH_4^+^_ and organic overloads that reduce nitrification activity, the need to control oxygen levels, and other shortcomings. In recent years, a bio-electrochemical system (BES), an emerging environmental biotechnology, has been used to treat nitrogen-containing wastewater [110]. In order to remove nitrogen from wastewater, the technique that utilizes microorganisms to catalyze the redox reaction of nitrogen oxides is used [111]. The process has a high denitrification efficiency and a clean denitrification process while improving the growth rate and activities of microorganisms in the system and actualizing energy recovery.

## 3. Nitrogen Recovery Process

Current processes for removing organic matter and nutrients are linear (production-consumption-disposal). As a result, these processes are unsustainable because they arise from converting reactive compounds produced using energy, either naturally (such as photosynthesis) or by industrial processes (such as ammonia synthesis), into non-reactive and benign forms. Resource recovery from wastewater will change the current situation under the constraints of emission standards. Various methods of recovering nitrogen (as ammonia) have been studied locally and globally. Still, nitrogen recovery remains complicated due to the absence of ready-made precipitates and the lack of investment in relevant technologies. However, several studies have shown [112] that ammonia can be efficiently recovered using membrane technology, which rejects other pollutants and produces viable fertilizer by-products. Current nitrogen recovery processes include ammonia stripping, MAP precipitation, and membrane technology. Their characteristics are compared in Table 3.

Membranes are materials with selective separation capabilities. Membrane separation involves the selective separating, purifying, and concentrating of various components of liquid mixtures using a membrane. It differs from traditional filtration in the following ways: membrane separation can occur at the molecular level; it can be operated at ambient temperature without a phase change; concentration and separation are performed concurrently; addition of other substances is not needed; the properties of the separated substances are unchanged; it is an adaptable, strong, and stable operation. The standard membrane processes for recovering NH_4^+^_-N include membrane distillation (MD), forward osmosis (FO), reverse osmosis (RO), nanofiltration (NF), and using anaerobic membrane bioreactors.

### 3.1. Membrane Distillation

Highly tolerant of salts, MD is driven by a temperature difference-induced transmembrane vapor pressure gradient, leading to the collection of transported vapor on the permeate side [115]. There are four basic MD configurations. Due to its inherent simplicity, direct contact membrane distillation (DCMD) has been the most studied configuration. Furthermore, vacuum membrane distillation (VMD) can be used for high yield. In contrast, air-gap membrane distillation (AGMD) and swept gas membrane distillation (SGMD) have the advantages of low energy loss and a high-performance ratio [116]. Table 4 details the advantages and disadvantages of these membrane distillation configurations [117,118].

In recent years, MD has been gradually applied to treat high ammoniacal nitrogen wastewater. Especially for DCMD, compared with other membrane distillation methods, DCMD using acid as the receiving solution has the advantages of higher selectivity, moderate ammonia flux, easy construction, and no additional cost [126]. For instance, Jacob et al. [127] used DCMD to treat livestock and poultry breeding wastewater. The ammoniacal nitrogen removal was >90%, and the anaerobic digestion effluent reached 99% after acidification pretreatment. In addition, Zico et al. [128] used a solar-heated MD system to treat landfill leachate with an ammoniacal nitrogen concentration of 2547 mg/L. They reported 98% ammonia removal and 59% recovery.

The pH of the feed liquid is a primary factor affecting NH_3_ separation. When the pH of the feed liquid increases, the overall mass transfer coefficient and separation factor will increase, increasing NH_3_ separation. He et al. [129] used NaOH to adjust the pH of the biogas slurry from 7.83 to 12.83 in ammonia recovery from swamps. At pH 10.0, the total mass transfer coefficient increased four times (from 0.36 × 10^−6^ to 1.53 × 10^−6^ m/s), and the separation factor increased nine times (from 6.61 to 60.3). The 95% total ammonia nitrogen is converted to free ammonia.

Similarly, the feed temperature is also an essential factor influencing MD. For instance, by increasing the temperature from 30 °C to 50 °C, Qu et al. [130] increased the permeate flux of NH_3_ removal by 250%. As the feed temperature increases, the feed ammonia concentration decreases more rapidly. When the feed temperature was increased from 30 °C to 55 °C, the Ka rose from 3.42 × 10^−5^ to 7.28 × 10^−5^ m/s. This rise occurred because the high feed temperature enhanced NH_3_ diffusion in the bulk solution and membrane pores, resulting in a higher mass transfer coefficient. Furthermore, the feed flow rate also affects the NH_3_ removal efficiency. Elsewhere, El-Bourawi et al. [131] achieved NH_3_ removal efficiencies of 55%, 60%, 63%, and 73% at feed flow rates of 0.28, 0.5, 0.75, and 0.84 m/s, respectively. As the feed flow rate increased, the NH_3_ removal efficiency increased gradually, attributable to concentration and temperature polarization effects. Generally, higher flow rates promote more turbulence, especially in the feed boundary layer near the membrane surface. In turn, it enhances heat and mass transfer from the bulk feed to the membrane surface, leading to higher NH_3_ removal.

However, in the NH_3_ recovery process, the chemical consumption cost accounts for more than 48% of the operating cost [114]. Therefore, Shi et al. [132] proposed a novel, two-stage membrane distillation (2s-MD) process. In its first stage, ammonium phosphate solution (MAP, NH_4_H_2_PO_4_) is used as a recoverable NH_3_ absorbent for NH_3_ recovery. Then, green NH_3_—diammonium phosphate solution (DAP, (NH_4_)_2_HPO_4_)—was recovered by vacuum distillation from the NH_3_-saturated MAP. The results showed this multi-cycle NH_3_ absorption-desorption experiment could recycle MAP/DAP and concentrate the regenerable NH_3_ water at 1.05 mol/L. Having excellent and stable quality, the recovered water can be reused for industrial or agricultural purposes.

### 3.2. Forward Osmosis (FO)

The FO process is based on the osmotic pressure difference between the feed solution (high water chemical potential) and the draw solution (low water chemical potential), driving the transport of ions across the membrane from the low-concentration solution (feed) to the high-concentration solution [133]. Generally, as water in the feed solution is drawn through the FO membrane, NH_4^+^_ is concentrated on the feed side. The FO principle is depicted in Figure 5.

FO membranes usually contain active and porous support layers [134]. In recent years, numerous experimental studies [135] have evaluated the diammonium performance of FO technology. Volpin et al. [136] used this process to recover 40–65% nitrogen from diluted human urine. Similarly, Zhang et al. [137] achieved 50–80% NH_4^+^_ removal from hydrolyzed urine using FO. 

The pH of the feed liquid side (FS) and the draw liquid side (DS) is one of the factors affecting denitrification via FO. When Engelhardt et al. [138] reduced the pH of the FS end from 6 to 3, the NH_4^+^_ rejection increased from 52% to >90%. Yet, the FO membrane and the draw solutes (DSs) are more essential factors in FO technology. Specifically, the characteristics of the FO membrane determine its applicability and efficiency [139]. Gonzales et al. [140] introduced positively charged polyethyleneimine (PEI) to the surface of an active polyamide layer to prepare positively charged thin film composite (TFC) membranes. Then, they used 1-ethyl-3-(3-dimethylaminopropyl) carbodiimide-mediated coupling with *N*-hydroxy succinimide, after grafting with PEI, to quaternize with iodomethane, forming quaternary amine groups on the TFC membrane. The results show that the amine grafting (98% NH_4^+^_ rejection) and quaternization (99% NH_4^+^_ removal) on the TFC membrane can enrich NH_4^+^_ in wastewater during the FO operation.

Different DSs, such as sodium chloride (NaCl) and magnesium chloride (MgCl_2_), have been widely used in FO. The DS characteristics affect FO efficiency. The ideal DS should have excellent osmotic pressure. Furthermore, it should have low viscosity, low reverse salt flux, a high diffusion coefficient, and low cost [141]. Experimenting with various DSs, Gulied et al. [142] showed that a single fertilizer draw solution (SFDS) produced a higher water flux than a multi-component draw solution (MCDS). The water fluxes of the SFDS containing KCl and NH_4_Cl were the highest, i.e., 4.45 and 4.43 L/m^2^·h, respectively, while the lowest, 3.32 and 2.63 L/m^2^·h were obtained by those with (NH_4_)_2_SO_4_ and NaH_2_PO_4_, respectively. The researchers inferred that the osmotic pressure generated by the SFDS was higher than that of the MCDS, resulting in an increased water draw and recovery. Likewise, Almoalimi et al. [143] evaluated the effects of different DSs on diammonium removal. The DSs studied included NaCl, MgCl_2_, MgSO_4_, Na_2_SO_4_, and non-ionic DSs such as glucose, glycine, and ethanol. The experimental results showed that for ionic DSs, the denitrification rates of those of NaCl, MgCl_2_, Na_2_SO_4_, and MgSO_4_ were 32.46%, 70.72%, 36.17%, and 89.5%, respectively. Still, MgSO_4_ with a higher diammonium removal efficiency had a water flux of only 8 L/m^2^ h. Therefore, it is tedious to simultaneously find ionic DSs with high water flux and high NH_4^+^_ removal efficiency. While all neutral chemicals used in this study achieved almost 100% NH_4^+^_ removal, selecting the neutral chemical with the most increased water flux was only necessary as DS. Considering the cost of glycine, glucose has the advantages of low price and easy availability; hence, glucose is a more suitable choice as DS for diammonium treatment.

However, the FO process still faces the following three problems: (1) high energy consumption and cost, caused by DS re-concentration; (2) most studies on FO membranes are still in the experimental (lab and pilot) phase; (3) low cleaning efficiency, and hard to be reused. Therefore, further studies on low-pollution operation, FO treatment of actual wastewater, and membrane cleaning strategies [144] are still important avenues for implementing FO technology to remove/recover NH_4^+^_.

### 3.3. Reverse Osmosis and Nanofiltration (RO and NF)

A membrane is, in general, a selective barrier between two phases. When a driving force is introduced into the system, permeation occurs [145]. The driving force is typically a difference in chemical potential caused by a pressure or concentration gradient across the membrane. Depending on membrane porosity and the trans-membrane gradient involved, a number of membrane processes appear, such as RO, NF, ultrafiltration (UF), and microfiltration (MF). RO is generally accepted to be a dense membrane material with a nonporous structure, while UF and MF are well recognized as porous membranes [146]. It is somewhat arbitrary to claim that NF, which is defined as a process between UF and RO that can separate multivalent ions from monovalent ions, is either porous or nonporous, because the openness of NF lies in the spectrum between discrete pores (UF/MF) and dense materials (RO) [147]. RO is a method of separating dissolved solids from wastewater using a selective osmosis membrane. In the RO process, the wastewater stream is divided into permeate and concentrated streams by additional pressure greater than the osmotic pressure and then discharged continuously. The osmotic pressure differential drives the influent from diluted to focused solutions through a semi-permeable membrane [148]. Figure 6 is a schematic diagram of the RO principle.

RO and NF were originally developed for the purification and recovery of drinking water from brine and brackish water. In recent years, both have gradually been used to remove pollutants such as heavy metals, nitrogen, and phosphorus [149,150,151]. Compared to other membrane technologies, they are more flexible, easy to operate, and relatively easy to maintain [152]. In addition, they are more economically feasible because they do not require a heat source or additional chemicals and do not produce phase transitions [153]. As a result, they are quickly becoming a standard technology for water purification in both the public and private sectors. Under certain operating conditions, RO removes > 99% of organic macromolecules and all microorganisms in wastewater [154]. Häyrynen et al. [155] investigated the recovery of NH_4^+^_ and nitrate from mine water with four commercial thin-film composite RO membranes and one NF membrane. They reported that the RO membranes produced high-quality permeate and concentrate for bioreactor treatment, while the NF membranes’ performance was poor. Also, the removal of NH_4^+^_ and nitrate in the concentration process of the RO membrane exceeds 82% and 90%, respectively. In another study, Moresi et al. [156] recovered ammonium fumarate from model solutions via NF and RO simulations. Their result showed the feasibility of removing microbial metabolites (such as ammonium fumarate) from model aqueous solutions using NF and RO. Similarly, Coskun [157] used NF and RO membranes to reduce the chemical oxygen demand (COD) concentration of olive factory wastewater from 40.0 to 1.00 mg/L at a pressure of 25 bar, achieving a combined COD removal efficiency of 97.5%. 

Furthermore, temperature is a crucial factor affecting two of the membranes’ performances. Theoretically, when the feed temperature increases, the diffusion coefficient increases and the viscosity decreases. According to Darcy’s Law, an increase in temperature increases the permeate flux due to a decrease in viscosity [158]. However, elevated temperature enhances concentration polarization and fouling because of the build-up of a fouling layer on the membrane, thereby reducing membrane efficiency. In order to optimize the membrane fouling of the integrated membrane process of NF and RO membranes, Kaya et al. [159] compared the effects of different temperatures on membrane fouling in the NF process to improve the permeation quality of the NF membrane. The results showed that membrane fouling gradually intensifies with increasing temperatures. Ray et al. [160] used RO, NF, and microfiltration (MF) to study the retention of urea and ammonia. The results showed that NF and RO recovered 90% and 64% of the non-ionic ammonia, respectively. For the hydrolysis of urine, the optimal recovery condition was the NF membrane with pH 11.5, recovering 90% of the non-ionic ammonia, 86% conductivity reduction, and 98% total organic carbon (TOC) rejection. Whereas for fresh urine, the optimal recovery conditions for RO and NF were pH 6 or higher. Also, Cancino-Madariaga et al. [161] evaluated the NH_4^+^_ recovery performance when the transmembrane pressure (TMP) in NF and RO membranes was changed. The NH_4^+^_ rejection at pH 5 and 7 was tested, and the experimental results showed that the maximum TMP values of the NF and RO membranes were 16 and 24.5 bars, respectively. NH_4^+^_ recovery at pH 7 was higher for all membranes evaluated than at pH 5 until the critical TMP was reached.

However, the RO/NF membrane still produces a large amount of concentrate during the separation and removal of nitrogen, requiring further treatment. When the wastewater contains suspended solids, the RO membrane efficiency is hindered [162], so the pretreatment separation process should prevent solids from being carried to the membrane system. Another disadvantage of using RO to treat wastewater is membrane fouling. Membrane fouling is mainly divided into four types, namely: colloidal particle fouling, inorganic fouling, organic fouling, and biological fouling [163]. Most of the first three types of fouling can be removed using specific cleaning methods, but biological fouling is extremely difficult to remove [164,165]. Biological contamination is uncontrollable due to the issue of biological activity. Even if the abundance of microorganisms in water is low after pretreatment, the microorganisms rapidly occupy the entire surface of a membrane through self-reproduction once they are adsorbed on the membrane. The extracellular polymeric substances (EPSs), which are secreted during reproduction and growth of microorganisms, combine with microorganisms to form a stable and difficult-to-degrade protective biofilm that is capable of effectively resisting the influence of the external environment [166]. Thus, biological fouling removal becomes complicated and difficult. The formation of such biofilms results in an increase in the water resistance of the membranes and a significant decrease in the water flux [167]. When the biofilm reaches a certain thickness, the separation membrane will be scrapped. Thus, how to effectively solve biological fouling has become an increasingly important topic. For instance, Epsztein [168] et al. used a NF-RO hybrid membrane filtration system, using NF membrane as a pretreatment of RO membrane to reduce membrane fouling. The results showed that this technology is achievable and can also achieve high water recovery. 

### 3.4. Anaerobic Membrane Bioreactor (AnMBR)

AnMBR is an integrated system that combines anaerobic bioreactors with membrane filtration. Adding membrane modules helps separate HRT from SRT, expanding its applicability to treating wastewater containing various constituents with a wide concentration range [169]. AnMBR is mainly built in two configurations [170]: lateral flow or submerged (Figure 7). 

In lateral flow mode, the membrane is placed outside the bioreactor, and the mixed liquor is transferred to the membrane to produce permeates. In contrast, the immersion mode of the AnMBR submerges the membrane in the main bioreactor. The configuration of the AnMBR can also be used as a two-stage system. This unique arrangement facilitates membrane cleaning and maintenance without affecting the microbial community in the main bioreactor. The retentate can be recycled from the membrane tank into the anaerobic tank to biodegrade the contaminants further and maintain the desired mixed liquid suspension (MLSS) in the tank.

In recent years, attention paid to AnMBR technology has been attributed to several unique AnMBR advantages, including a smaller footprint, no solid infiltration, and less sludge production. It is due to the anaerobic environment with relatively low excess biosolid collection and disposal costs. In addition, the process converts organic matter to methane for energy recovery, minimizing operational energy requirements and accomplishing high COD removal, among others [171]. To date, this process has been used for many wastewater treatments. For instance, Grossman et al. [172] used the method to recover 77% nitrogen and 57% TOC from food processing wastewater and methane, respectively. Also, 91% of the phosphorus was simultaneously recovered. Elsewhere, Kong et al. [173] used a large-scale, submerged AnMBR to treat urban sewage at 25 °C. They reported > 90% COD removal and >95% biological oxygen demand removal under low HRT for 6 h. 60–64% of influent COD (>250 mg/L) is converted to CH4, 20–26% of COD is converted to MLSS, and MLSS absorbs 10–16% of influent nitrogen.

Since an anaerobic membrane bioreactor requires NOB growth, temperature, C/N, and SRT, they affect the nitrogen removal efficiency of an AnMBR. Specifically, temperature directly influences the activated sludge’s physical, chemical, and biological properties, especially under anaerobic conditions. Gao et al. [174] used AnMBR to study the effect of temperature on microbial communities and membrane biofouling. The results showed that the COD removal efficiencies were 85%, 93%, and 90% at 15 °C, 25 °C, and 35 °C, respectively. In addition, the low-temperature condition facilitates membrane fouling more than the mesothermal state (the membrane fouling cycle changes from 23 to 8 days when the temperature decreases from 35 °C to 15 °C). The influent C/N is also one of the most influential parameters affecting denitrification, as insufficient carbon sources limit the efficiency of the denitrification process [175].

Furthermore, a low C/N may adversely affect membrane fouling because it alters microorganisms’ physiological properties and the biomass’s chemical composition in membrane bioreactor (MBR) [176]. Chen et al. [177] studied the effect of C/N on SAAMB-AnGMBR performance. They observed that when the C/N was 100/5, the nitrogen removal rate exceeded 91%. However, when the values dropped to 100/8 and 100/10, the nitrogen removal reduced substantially by 68.9% and 44.1%, respectively. The results also show that lowering the C/N balance significantly retards TN removal, worsening particle quality, and aggravating pore clogging, which, in turn, aggravates membrane fouling.

SRT is also a crucial factor; when excessively short, it can prevent the hydrolysis of organic matter while promoting membrane fouling. Whereas, when overly long, it would increase sludge viscosity, rapidly increasing the membrane surface resistance, resulting in a significant decrease in permeate flux [178]. Using this method with SRT conditions for 30, 60, and 90 days, Huang et al. [179] found that the 90-day SRT benefits biomass accumulation, organic biodegradation, and methane production. As the SRT increased from 30 to 90 days, the hydrolysis of the organic matter increased from 35% to 56%. Also, the membrane fouling of the 30-day SRT rose from 35% to 56%. This membrane fouling was the fastest, followed by that of the 90-day SRT before the 60-day SRT. The trend is consistent with the literature above [178].

However, membrane fouling in the AnMBR is more severe than in other modalities due to the higher content of microbial products [180]. While membrane fouling mitigation via physical, mechanical, chemical, and biological methods has been adequately researched, optimizing strategies to control membrane fouling is still ongoing [181]. Nearly, there have been studies biological carriers (such as sponges), plastic media, and granular media (such as granular activated carbon (GAC) are added to the MBR to form hybrid membrane bioreactors (MBRs), which significantly mitigate membrane fouling [182].

Similarly, Wang et al. [183] evaluated the addition of polyvinyl formal (PVFM) bio-carriers to membrane bioreactors at high and low C/N values of 20.0 and 6.7. The results showed that adding bio-carriers could enrich nitrifying and denitrifying bacteria and slightly increase the total nitrogen removal rate, even at lower C/N values. On the other hand, developing new biofilm reactors shows excellent potential for reducing energy demand and improving performance, especially the anaerobic membrane distillation bioreactor (AnMDBR). In the anaerobic osmotic membrane bioreactor (AnOMBR) studied in recent years [184,185], integrated MD and FO processes with the AnMBR enhanced water recovery and pollutant removal capacity.

## 4. Conclusions

This article reviews several standard processes for denitrification and the recovery of nitrogen. The review includes the advantages and disadvantages of the different techniques and the factors that affect each process. Among them, biological methods seem to be the most efficient treatment processes, with >90% removal efficiency. Also, the denitrification performance of this process is mainly affected by pH, temperature, sludge retention time, carbon-nitrogen ratio (C/N), hydraulic retention time, etc. 

MAP precipitation has been trendy in wastewater denitrification in recent years. It is attributed to its excellent economy, simple and convenient operation, lack of pipelines and other transportation equipment blockages, and simultaneous recovery of NH_4^+^_ salt and orthophosphate. It is mainly affected by pH, Mg^2+^:NH_4^+^_:PO_4_^3−^, and the choice of magnesium source. 

On the other hand, adsorption is also popularly used in wastewater denitrification. Its advantages include high removal efficiency, convenient operation, and low energy consumption. It is especially suitable for treating wastewater with low NH_4^+^_ concentrations. The limiting factors for its denitrification performance include pH, contact time, the surface ratio between the adsorbent and adsorbate, and metal ions in the wastewater. Among the three main denitrification methods, the biological method is more practical for wastewater denitrification due to the following advantages: (1) high denitrification efficiency, which can remove most of the organic nitrogen and ammonia nitrogen in sewage; (2) simple process, low investment, and low cost; (3) simultaneous removal of multiple pollutants, which can degrade organic compounds such as phenol, cyanide, and COD while denitrifying.

Membrane technology has been the primary nitrogen recovery process in the past few years. It has replaced the traditional activated sludge treatment of wastewater. Different membrane methods are available to treat wastewater based on the need to provide better effluent quality while lowering the cost. Among the several membrane technologies for recovering nitrogen described above, the membrane bioreactor has the best application prospects. Compared with other membrane recovery technologies, the membrane bioreactor has the following advantages: (1) efficient solid-liquid separation and good water quality; (2) it can realize the complete separation of HRT and SRT, and the operation control is flexible and stable; (3) it is beneficial to the interception and reproduction of nitrifying bacteria, and the nitrification efficiency of the system is high. It can also have the functions of deaminization and dephosphorization by changing the operation mode. 

However, these nitrogen treatment processes still have some room for improvement. For instance, the high operating costs of biological methods to dispose of unwanted sludge render the methods economically unfriendly. As a result, several control strategies (such as DO control, C/N control, etc.) are integrated and researched to find an optimal option that can cover most influencing parameters, reduce the start-up time, and improve efficiency. 

Furthermore, chemical precipitation exhibits the lowest denitrification efficiency (20–80%). It is highly pH-dependent. Therefore, it can be considered to combine electrochemical methods with MAP precipitation. Electrochemistry can efficiently remove P and N and convert P and N into precipitates and other products. Specifically, in electrochemical methods using Mg as the anode, the electrochemical dissolution of magnesium continues as long as the compound is continuously in contact with the aqueous solution. Continuous dissolution provides additional Mg^2+^ to promote MAP crystal growth. The adsorption method primarily depends on finding a suitable adsorbent. Therefore, identifying a novel solid media or modifying existing ones, remains a significant research direction dilemma.

Although membrane technology has made some progress in wastewater treatment, it still faces some challenges. They include unavoidable membrane fouling and low energy recovery efficiency. Therefore, conducting in-depth research on fouling control and membrane material or membrane module design to control membrane fouling and reduce energy requirements is imperative. Also, researching new mechanical cleaning and flushing methods to achieve effective pollution control and new bottom membranes and adding different active layers to improve denitrification performance and reduce energy consumption are two promising research directions.

Considering the complexity of wastewater, the peculiar advantages and disadvantages of each separation technique, and the desired water quality and effluent characteristics, combining various methods to treat wastewater efficiently and economically should be encouraged. 

## Figures and Tables

**Figure 1 ijerph-20-03429-f001:**
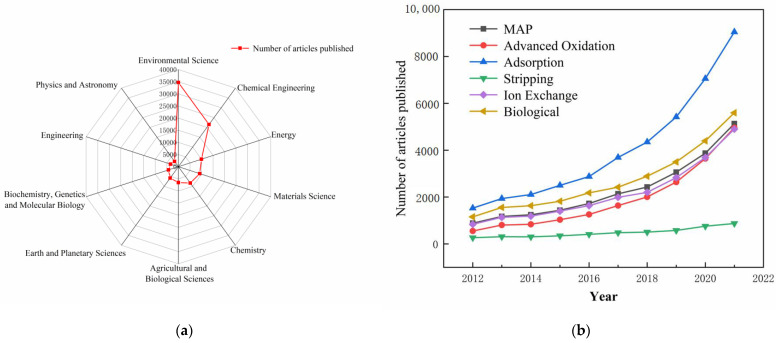
Literature survey on wastewater denitrification from 2012 to 2021: (**a**) fields of the relevant publications and (**b**) the degree of attention given to various denitrification methods used.

**Figure 2 ijerph-20-03429-f002:**
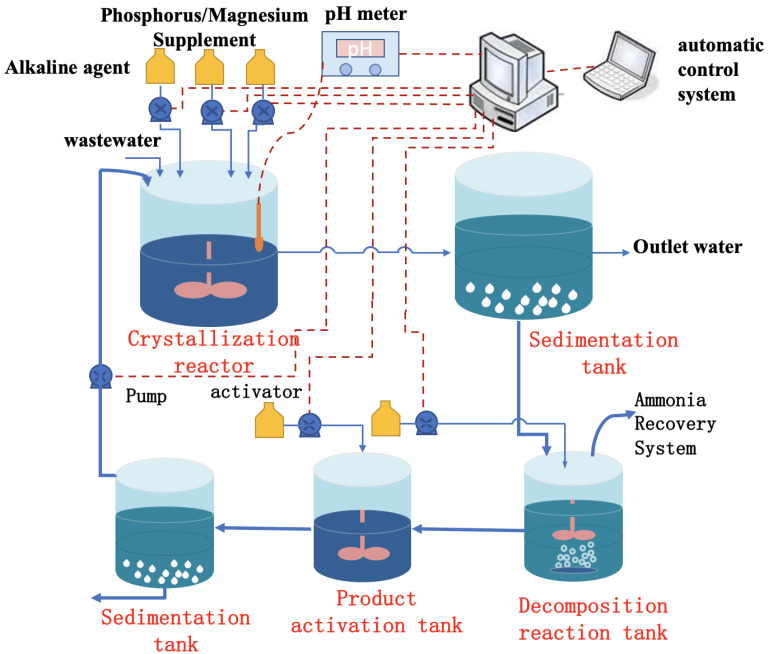
Schematic of the denitrification process by ammonium magnesium phosphate precipitation.

**Figure 3 ijerph-20-03429-f003:**
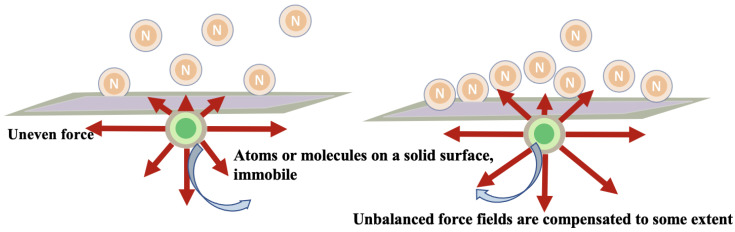
A schematic representation of the adsorption principle.

**Figure 4 ijerph-20-03429-f004:**
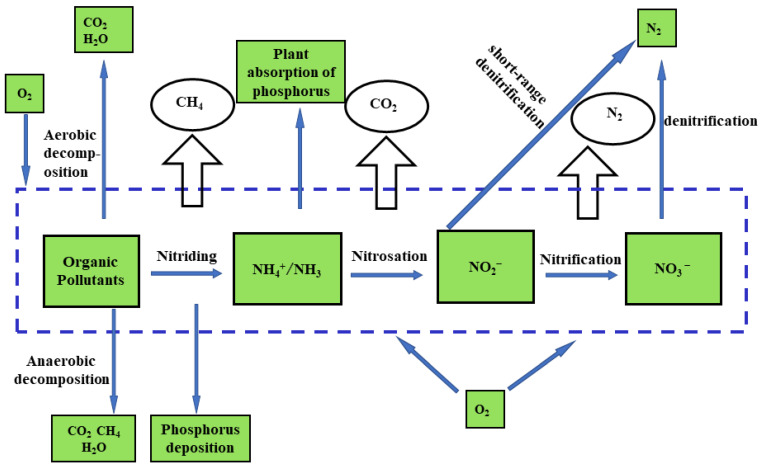
A schematic showing the general steps of biological denitrification in wastewater treatment.

**Figure 5 ijerph-20-03429-f005:**
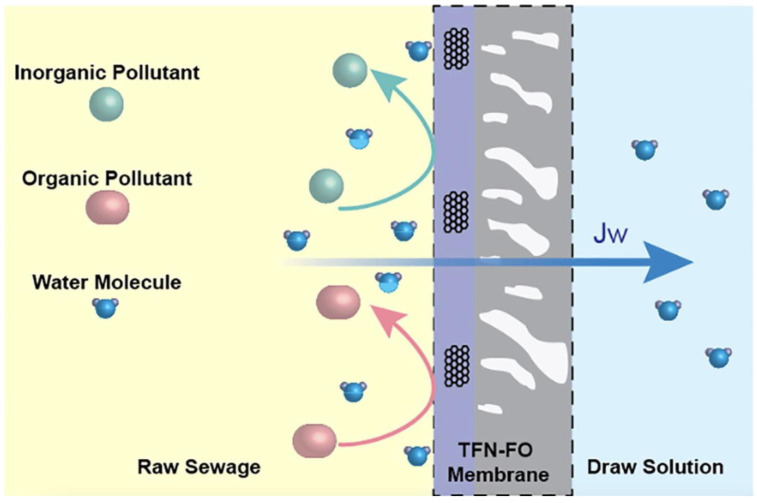
Schematic of forward osmosis (FO) principle (reprinted with permission from Ref. [133] Wu et al., 2021).

**Figure 6 ijerph-20-03429-f006:**
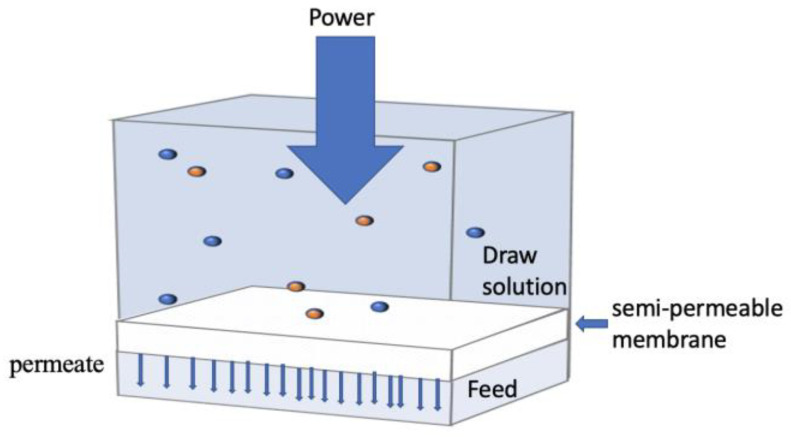
RO membrane process principle.

**Figure 7 ijerph-20-03429-f007:**
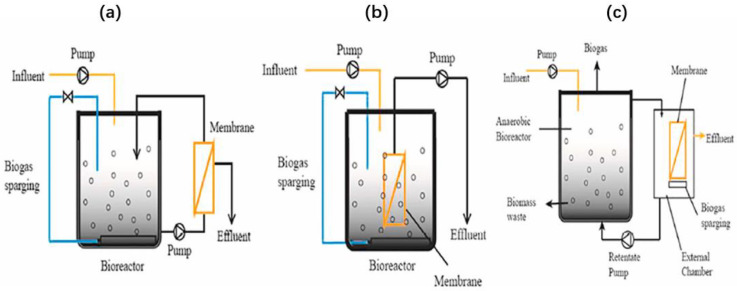
Basic configuration of an anaerobic membrane bioreactor: (**a**) flow measurement; (**b**) internal submerged; (**c**) external submerged (Reprinted with permission from Ref. [170] Al-Asheh et al., 2021).

**Table 1 ijerph-20-03429-t001:** Standard technologies for removing NH_3_/NH_4^+^_ from wastewater.

Method	Working Environment	Advantage	Shortcoming	Outlet Concentration (mg/L NH_4_–N)	Removal Efficiency
Chemical precipitation [41,42]	Requires a specific pH and temperature	Produces valuable fertilizers at a moderate cost	Requires additional magnesium source; incurs phosphate cost; introduces new contaminants	29–100	20–98%
Adsorption method [43,44]	Broad temperature and pH range	Simple and effective removal of NH_4^+^_; able to work at low NH_4^+^_ concentrations	Adsorbents have different removal efficiencies	1	43–97%
Biological method [29,45]	Heterotrophic, photosynthetic algal, or bacterial growth is temperature sensitive	No need for chemical reagents and complicated configurations; high denitrification efficiency	High cost; requires external carbon source; only operates at low input/output concentrations; long start-up time	<5	70–99.9%

**Table 2 ijerph-20-03429-t002:** Comparison of various indicators among various denitrification and conventional treatment processes.

Operational Technology	Total Nitrogen Removal Efficiency	Cost	Effect	Main Parameters
Partial nitrification of nitrite [96]	The nitrite reduction rate is increased by 1.5–2 times in the subsequent denitrification stage	40% reduction in COD ^1^	25% reduction in oxygen demand and 20% reduction in CO_2_ emissions during denitrification	pH, temperature, DO ^2^, real-time aeration control, SRT ^3^, and substrate concentration
Simultaneous nitrification and denitrification [97,98]	Nitrogen removal rate 99%	Requires external carbon sources	Almost complete removal of organic matter and NH_4^+^_-N, no accumulation of by-products	DO, carbon source, reactor design, oxygen availability for nitrification, and efficient carbon source utilization for denitrification
Short-path nitrification and denitrification [99]	The nitrite denitrification rate is 1.5–2 times higher than the nitrate denitrification rate	40% reduction in electron donor requirement during the anaerobic phase	In the aerobic stage, the oxygen demand is reduced by 25%, and the energy saving is 60%. It is suitable for wastewater with high ammonia concentration or low C/N ratio	DO, HRT ^4^, pH, C/N ^5^ ratio, substrate concentration, and aeration mode
Simultaneous partial nitrification, anammox, and denitrification [100]	99% denitrification	Low concentration of organic matter	Simultaneous removal of inorganic nitrogen and organic carbon, suitable for wastewater with complex composition, high ammonia concentration, and low C/N ratio	Intermittent aeration, pH, DO, C/N ratio, and free ammonia concentration
Anammox [101]	The denitrification rate is over 90%	100% Reduction in organic carbon source requirements	Oxygen demand is reduced by 60%, and N_2_O production is reduced. In addition, the anammox process produces 90% less sludge, which reduces sludge disposal costs	Reactor configuration, initial biomass concentration, usually for high NH_4^+^_ ion wastewater

COD ^1^ = chemical oxygen demand; DO ^2^ = dissolved oxygen; SRT ^3^ = sludge retention time; HRT ^4^ = hydraulic retention time; C/N ^5^ = carbon to nitrogen ratio.

**Table 3 ijerph-20-03429-t003:** Indicators of nitrogen recovery processes.

Method	Principle	Nitrogen Removal Efficiency	By-Product	Shortcoming
Ammonia stripping [113]	Through the difference in gas partial pressure, free ammoniacal nitrogen escapes from wastewater in a gaseous state	50–98%	Ammonium sulfate	Large air consumption, high energy consumption, secondary pollution. Easy scaling,
MAP precipitation [41]	By adding chemical reagents to form precipitation to achieve solid-liquid separation, to separate ammoniacal nitrogen	65–98%	MAP precipitation	Requires additional phosphorus and magnesium sources, and causes secondary pollution
Membrane technology [114]	Separation of nitrogen gas by selective ion permeation through membranes	64–99.8%	Ammonium salt fertilizer	Membrane fouling and wastewater contaminants settle on the membrane surface, reducing the efficiency

**Table 4 ijerph-20-03429-t004:** Advantages and disadvantages of various types of membrane distillations.

Configure	Advantage	Shortcoming	Nitrogen Recovery Efficiency	Reference
DCMD	1. High permeation flux 2. Simple design and operation 3. The flux is more stable, and the output ratio is high	1. Low thermal efficiency 2. Temperature and concentration polarization have great influence 3. The water quality is seriously polluted.	>90%	[119,120,121]
VMD	1. High permeation flux 2. Less conduction heat loss 3. The water quality of the product is good	1. The wettability of membrane pores are strong, and the membrane fouling rate is higher. 2. Heat recovery is difficult, requiring a vacuum pump and an external condenser.	>95%	[122,123]
AGMD	1. Seawater can be used as the cooling water flow on the permeate side. 2. High thermal efficiency 3. Relatively high throughput	1. Further resistance to water vapor results in a lower permeate flux. 2. The air gap provides additional resistance to steam; the module is difficult to design	88%	[124]
SGMD	1. High mass transfer rate 2. Low heat loss by conduction	1. Heat recovery is difficult. 2. Handling sweep gas is difficult. 3. Larger condenser is required, and the cost is higher	85%	[125]

## Data Availability

Not applicable.

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
