# Peer review of "A Comprehensive Review on Wastewater Nitrogen Removal and Its Recovery Processes"

_ijerph, 2023, doi:10.3390/ijerph20043429_

Round 1

Reviewer 1 Report

This paper reviews three common types of denitrification processes and mainly focuses on the membrane technology for nitrogen recovery. The topic is interesting and is within the scope of the journal. In general, I recommend that the article needs minor revisions. Following are my detailed suggestions for future improvements, and then the same can be accepted.

1. It is well known that the membrane technology is an efficient and environmentally friendly method for nitrogen recovery, which many previously reported in several papers. However, the reference list is not up to date for wastewater nitrogen removal and its recovery processes. Several papers have been published in 2022 and 2023, please update the latest literature reviews. For example, Table 1 can be improved by adding several recently published papers.

2. Detailed explanations are not made when formulas are listed in the manuscript.

3. The figures in the manuscript are not particularly clear, such as Figure 7, which can be updated.

Author Response

请参考附件。

Reviewer 2 Report

This paper review three different types of denitrification processes, including physical, chemical, and biological processes, and mainly focuses on recovering of Nitrogen from wastewater using membrane technology. The paper could be published in IJERPH after major revisions. 

1. In the abstract, On line 14, Please remove the space (Finally, it is proposed .... etc)

2. On line 155, should be Figure 3 not Figure 1

3.  The authors can refer to the recent paper in the field of nitrogen recovery (like https://doi.org/10.3390/membranes13010015) to enrich the review content for the broad readers. 

4. On line 460, Page 14, more information needs to be included about membrane fouling in RO membranes and how to combat with this problem

5. Is RO membranes porous or nonporous and why??? Can the authors write few sentences about nanofiltration (NF). In section 3.3, just RO is explained in detail. Thus, more information is to be included. 

6. On line 322, Page 10, the authors presented the advantages and disadvantages of different types of MD membranes, it would be better to compare the advantages and disadvantages in terms of nitrogen recovery. I have just seen and read general information included in the Table 4

7. In MD modes, which mode is better for nitrogen recovery and why. This needs to be included in MD sections

8. In Conclusion section, which types of membranes (MD, FO, RO, NF and AnMBR) the best for nitrogen recovery from wastewater.

9. The abbreviations should be supplemented. 

10. What TOC, BOD, MLSS, TFC, NOB stands for?

Reviewer 3 Report

Comments to the author:

Research article entitled “A Comprehensive Review on Wastewater Nitrogen Removal and Its Recovery Processes” reviewed on the three common types of denitrification processes, including physical, chemical, and biological processes, and mainly focuses on the membrane technology for nitrogen recovery. Authors focused on an interesting topic. Manuscript can be accepted for publication in “Int. J. Environ. Res. Public Health” after addressing the following comments.

1.      In the introduction authors should discuss on the currently existing reviews on the Nitrogen Removal and Its Recovery Processes

2.      Currently on-site sensing strategies are emerging. Some potential literature is dedicated on the on-site detection of toxic pollutants and hazardous constituents that can provide a basic idea; “Colorimetric based on-site sensing strategies for the rapid detection of pesticides in agricultural foods: New horizons, perspectives, and challenges”; “Emerging insights into the use of carbon-based nanomaterials for the electrochemical detection of heavy metal ions” and “Portable electrochemical sensing methodologies for on-site detection of pesticide residues in fruits and vegetables”

3.      Authors should discuss on the challenges that are currently facing with the Nitrogen Removal and Its Recovery Processes

4.      Currently various technologies are available for the Nitrogen Removal and Its Recovery Processes; Among these technologies; which one is the best for the removal of nitrogen from waste water.

Round 2

Reviewer 2 Report

Thanks for your efforts in modifying the manuscript. 

Reviewer 3 Report

Accept